# Secreted Soluble Factors from Tumor-Activated Mesenchymal Stromal Cells Confer Platinum Chemoresistance to Ovarian Cancer Cells

**DOI:** 10.3390/ijms24097730

**Published:** 2023-04-23

**Authors:** Yifat Koren Carmi, Hazem Khamaisi, Rina Adawi, Eden Noyman, Jacob Gopas, Jamal Mahajna

**Affiliations:** 1Department of Nutrition and Natural Products, Migal–Galilee Research Institute, Kiryat Shmona 11016, Israel; 2Shraga Segal Department of Microbiology, Immunology, and Genetics, Ben-Gurion University of the Negev, Beer Sheva 8400101, Israel; 3Department of Oncology, Soroka University Medical Center, Beer Sheva 8400101, Israel; 4Department of Biotechnology, Tel Hai College, Kiryat Shmona 1220800, Israel

**Keywords:** chemoresistance, HGF, IL-6, kinase inhibitors, ovarian cancer, tumor microenvironment

## Abstract

Ovarian cancer (OC) ranks as the second most common type of gynecological malignancy, has poor survival rates, and is frequently diagnosed at an advanced stage. Platinum-based chemotherapy, such as carboplatin, represents the standard-of-care for OC. However, toxicity and acquired resistance to therapy have proven challenging for the treatment of patients. Chemoresistance, a principal obstacle to durable response in OC patients, is attributed to alterations within the cancer cells, and it can also be mediated by the tumor microenvironment (TME). In this study, we report that conditioned medium (CM) derived from murine and human stromal cells, MS-5 and HS-5, respectively, and tumor-activated HS-5, was active in conferring platinum chemoresistance to OC cells. Moreover, CM derived from differentiated murine pre-adipocyte (3T3-L1), but not undifferentiated pre-adipocyte cells, confers platinum chemoresistance to OC cells. Interestingly, CM derived from tumor-activated HS-5 was more effective in conferring chemoresistance than was CM derived from HS-5 cells. Various OC cells exhibit variable sensitivity to CM activity. Exploring CM content revealed the enrichment of a number of soluble factors in the tumor-activated HS-5, such as soluble uPAR (SuPAR), IL-6, and hepatocyte growth factor (HGF). FDA-approved JAK inhibitors were mildly effective in restoring platinum sensitivity in two of the three OC cell lines in the presence of CM. Moreover, Crizotinib, an ALK and c-MET inhibitor, in combination with platinum, blocked HGF’s ability to promote platinum resistance and to restore platinum sensitivity to OC cells. Finally, exposure to 2-hydroxyestardiol (2HE2) was effective in restoring platinum sensitivity to OC cells exposed to CM. Our results showed the significance of soluble factors found in TME in promoting platinum chemoresistance and the potential of combination therapy to restore chemosensitivity to OC cells.

## 1. Introduction

Ovarian cancer (OC) ranks as the second most common type of gynecological malignancy, and it has poor survival rates [1]. While platinum-based chemotherapy constitutes the standard-of-care for OC, toxicity and acquired resistance have proven challenging in the treatment of patients with OC [2,3]. Chemoresistance is frequently encountered in OC patients, consequent to multiple mechanisms, including decreased platinum accumulation within the cancer cells, elevated drug inactivation by metallothionein and glutathione, and enhanced DNA-repair activity [4,5]. In addition to chemoresistance, resulting from alterations within the cancer cells [6,7], an increasing number of studies have also implicated the tumor microenvironment (TME) in cancer chemoresistance [8,9].

The tumor microenvironment consists of a number of cell types, including immune cells, fibroblasts, adipocytes, and mesenchymal stem cells (MSCs). MSCs play a complicated role in cancer by differentiating and inducing pro- or anti-tumorigenic activity [10]. They home into the tumor site, following secretion of soluble factors, such as IL-6, IL-1β, and/or transforming growth factor-β1 (TGF-β1) by tumor cells, as well as the stromal cell-derived factor1α (SDF-1α) by stromal cells [10] that promote paracrine signaling in tumor cells. MSCs also migrate to the TME, consequent to hypoxia and inflammation [11,12]. It has been suggested that, within the tumor, the MSC undergoes a switch between anti- and pro-tumorigenic phenotypes, which dictate tumor progression [13,14].

TME cellular compartments interact with tumor cells via various mechanisms, including secretion of growth factors. In breast cancer, it has also been observed that interlukin -6 (IL-6), secreted by breast cancer-associated MSCs, protected cancer cells from cisplatin-induced apoptosis by activating the signal transducer and activator of the transcription 3 (STAT3) pathway [15].

MSCs are associated with TME-mediated chemoresistance by secreting soluble factors, such as stromal cell-derived factor 1 (SDF-1), IL-6, nitric oxide (NO), interlukin-3 (IL-3), G (granulocytes)-colony-stimulating factor (CSF), M (macrophages)-CSF, and GM-CSF, as well as the activation of proliferation pathways [16]. They also utilize autophagy to synthetize anti-apoptotic factors and extracellular matrix (ECM) that protect and induce tumor cells growth [17].

In OC, IL-6 secretion by MSC induced tumor cells’ expression of Bcl-2 and Bcl-XL, as well as consequent apoptosis inhibition. Moreover, exposure of MSC to cisplatin promoted the phosphorylation of WNK-1, c-Jun, STAT3, p53, and other tyrosine kinases. This, in turn, promoted MSC survival and IL-6 and IL-8 secretion, which affected cancer cells’ chemoresistance [14,18].

Previously, we demonstrated that direct co-culture of OC cells with MSC conferred chemoresistance to therapeutic agents, including paclitaxel, colchicine, and platinum compounds, accompanied by blocking of ERK1/2 activation [19]. We also demonstrated that the combination of a platinum drug with fisetin and other flavonoids restored platinum drug sensitivity to OC cells that were co-cultured with MSC. Restored platinum sensitivity was accompanied by re-activation of ERK1/2 [19]. Moreover, 2-hydroxyestardiol (2HE2), an estradiol metabolite that serves as a prodrug for 2-methoxyestradiol (2ME2) and converts efficiently to 2ME2, was also active in restoring platinum sensitivity to OC cells growing in direct contact with MSCs [20].

In this study, we utilized our cisplatin derivative, RJY13 [21], to explore the involvement of soluble factors secreted from MSCs and adipocyte cells in promoting OC chemoresistance. We report that CM derived from differentiated 3T3-L1, MS-5, HS-5, and tumor-activated HS-5 (TA HS-5) cells were active in conferring platinum chemoresistance to OC cells. Various OC cells exhibit variable sensitivity to CM activity. The tumor-activated HS-5 is enriched in soluble uPAR (SuPAR), IL-6, and hepatocyte growth factor (HGF), which might be responsible for chemoresistance induction. FDA-approved JAK inhibitors mildly restored platinum sensitivity to OC cell lines. Moreover, Crizotinib, an ALK and c-MET inhibitor, combined with a platinum compound, block HGF’s ability to promote platinum resistance and restoration of platinum sensitivity to OC cells. Finally, exposure to 2HE2 restores platinum sensitivity to OC cells exposed to CM. Our results show TME’s significance in acquired platinum chemoresistance and the potential of modulators, which regulate a variety of signaling pathways in the restoration of chemosensitivity to OC cells.

## 2. Results

### 2.1. Condition Media Derived from Mesenchymal Stem Cell (MSC) and Tumor-Activated MSC (TA MSC) Promote Chemoresistance in OC Cell Lines

Previously, we showed that co-culture (CC) of OC cells with MSC cells (human and murine) confers platinum chemoresistance as monitored by PARP cleavage assay [19]. In this study, we focused on secreted soluble factors and tested their ability to confer chemoresistance. Initially, we pre-treated OC cell lines (A2780, A2780cisR, and OV-90) with condition media (CM) collected from human MSC, or from tumor (OC cells)-activated MSC (co-culture of OC cells with HS-5), prior to the application of RJY13 and monitored viability of OC cells and apoptosis induction by PARP-cleavage assay [19].

Exposure of A2780cisR, a cisplatin-resistant OC cell line, to HS-5-derived CM, caused an increase in cell viability by 46% when cells were treated with RJY13. Moreover, live cell count was further increased in A2780cisR cells pre-treated with CM collected from TA HS-5 cells (85% versus untreated and platinum-treated cells). Increased cell viability of A2780CisR cells treated with RJY13 in presence of human MSC-derived CM was accompanied by reduced levels of apoptosis, as monitored by the decrease in cleaved PARP (cPARP) levels (Figure 1B,C). cPARP levels were lowered by 33% and 39%, when CM of HS-5 or CM of TA HS-5 was used, respectively. This indicates that partial RJY13 chemoresistance may be conferred by these CMs.

Similarly, TA MSC-derived CM also conferred chemoresistance to A2780, a cisplatin-sensitive OC cell line (Figure 1B). Interestingly, CM collected from HS-5 (human MSC) enhanced levels of cPARP twofold versus control. Furthermore, CM media derived from MS-5 (murine MSC) reduced cPARP levels by about 30%. In contrast, CM collected from TA MSCs (human or murine) reduced cPARP levels approximately threefold.

Finally, CM media derived from MSCs or TA MSCs were active in conferring platinum chemoresistance to OV-90 cell lines, a high-grade serous ovarian cancer histology type [22] (Figure 1C).

Our data suggest that considerable chemoresistance was conferred via condition media, secreted by MSCs or, more robustly, by TA MSC cells. However, various OC cells exhibit variable sensitivities to both types of CM.

### 2.2. ERK1/2 Involvement in Secreted Soluble Factor-Mediated Chemoresistance

CM’s ability to confer platinum chemoresistance to OC cells was not limited to MSCs. We monitored the ability of CM derived from adipocyte cells to affect platinum chemoresistance in OC cells.

The results shown in Figure 2A illustrate that CM derived from the progenitor 3T3-L1 were partially active in conferring platinum chemoresistance to A2780CisR cells. However, CM derived from differentiated 3T3-L1 adipocyte (WAT) were more potent in conferring platinum chemoresistance to OC cells (Figure 2A). Previously, we demonstrated that direct co-culture of MSCs with OC cells conferred platinum chemotherapeutic, which was dependent upon blocking of ERK1/2 activity [19]. The flavonoids fisetin and quercetin were active in restoring ERK phosphorylation, as well as sensitivity to platinum compounds [19]. Based on these results, we further explored ERK1/2 signaling’s role in soluble factor-mediated chemoresistance.

To do so, we monitored phospho-ERK1/2 levels in OC cells that were stimulated upon platinum exposure. The ability of platinum exposure to stimulate phosphorylation of ERK1/2 was not significantly affected by the presence of CM derived from MS-5, 3T3-L1, or differentiated 3T3-L1 cells (Figure 2A). Moreover, pERK1/2 levels were also measured in the three OC cell lines tested–A2780cisR, A2780, and OV-90–following incubation with CM derived from HS-5 or TA HS-5 cells. In addition, fisetin, an ERK1/2 activator, was employed alongside the CM to examine its effect on sensitivity of OC to RJY13.

Again, exposure to platinum stimulated ERK1/2 phosphorylation in all cell lines tested (A2780cisR, A2780, and OV-90) (Figure 2B). Moreover, a reduction in pERK1/2 levels in samples exposed to CM derived from HS-5 or TA HS-5 (TA-HS-5) was correlated to decreased cPARP levels. However, as shown in Figure 2C, there is no clear correlation between pERK1/2 and RJY13 resistance. Only a limited reduction in pERK1/2 in A2780 cells treated with HS-5 CM was observed, while RJY13 sensitivity increased. In addition, TA HS-5-derived CM resulted in decreased pERK1/2, accompanied by RJY13 resistance (the same trend as observed for A2780cisR cells). In contrast, while OV-90 cells exhibited no change in ERK1/2 phosphorylation when cells were pre-treated with HS-5 CM, ERK1/2 activation was increased when pre-treated with TA HS-5 CM (Figure 2D). In all cases, significant RJY13 chemoresistance was conferred. CM incubation variably affected pERK1/2, depending upon the cell line tested and the source of the CM. Moreover, ERK1/2 phosphorylation does not correlate to platinum chemoresistance. This finding is in contrast to our previous finding that blocking of pERK1/2 phosphorylation correlates with drug resistance upon direct MSC interaction with OC cells [19]. Thus, platinum chemoresistance to OC cells that were mediated by exposure to CM likely is not significantly dependent upon ERK1/2 activity.

Exposure to fisetin increased the levels of ERK1/2 phosphorylation, but it did not result in cPARP cleavage following RJY-13 treatment (Figure 2E). Thus, there is no correlation between ERK1/2 phosphorylation and platinum sensitivity in OC cells exposed to CM derived from TA HS-5 versus ERK1/2 correlating with platinum chemoresistance in OC cells grown in direct co-culture with MSCs [19].

### 2.3. SuPAR, IL-6 and Hepatocyte Growth Factor Are Enriched in TA HS-5 Condition Medium

To identify soluble factors responsible for promoting chemoresistance in OC cell lines, we screened the three CM for cytokines, chemokines, and growth factors. First, we confirmed that the collected CMs are active in inducing chemoresistance (Figure 3A), followed by screening the three CMs using the commercial XL cytokine array kit (R&D Systems), which enables semi-quantitative determination of about 105 different cytokines (Figure 3B). Results shown in Figure 3D indicate that SuPAR (soluble receptor), IL-6, and hepatocyte growth factor (HGF) were enriched in the active CM, and they correlate with chemoresistance induction. CM collected from TA HS-5 and CM HS-5 monolayers were also tested using a multiplex ELISA kit for MCP-1, RANTES, leptin, IL-10, IL-8, IL-6, IL-1β, and TNFα presence. Among these, MCP-1 (CCL-2) and RANTES (CCL5) were detected in TA HS-5 CM relative to CM HS-5 (Figure 3E).

SuPAR is a soluble secreted receptor, able to deplete its ligand from the medium and prevent the initiation of its cellular signaling and prevent triggered signaling [23]. Thus, in this study, we focused on the IL-6 and HGF that were implicated as potential inducers of cellular signal transduction pathways, leading to chemoresistance [24].

### 2.4. JAK/STAT Inhibitors Partially Reduce RJY13 Chemoresistance in OC Cell Lines Pre-Treated with TA HS-5 CM

The activity of many of the soluble factors that were enriched in the active CM, such as IL-6, is mediated by the Janus kinases/signal transducer and activator of the transcription JAK/STAT pathway [25]. To validate the involvement of soluble factors that stimulate JAK/STAT activity in promoting platinum chemoresistance to OC cells, we used the following JAK inhibitors: Ruxolitinib, a FDA-approved JAK1 and JAK2 inhibitor [26]; and we used Tofacitinib, a JAK3 inhibitor [27]. These were added together with the TA HS-5 CM. In addition, we utilized Dasatinib, a Src and Abl inhibitor [28], which was previously found efficient in re-sensitizing cells to platinum drugs [29]. The results in Figure 4 show that partial restoration of RJY13 sensitivity was achieved in the presence of Ruxolitinib and Tofacitinib, as measured by mildly elevated cPARP band intensity in A2780 cells (Figure 4A). In contrast, when the platinum-resistant A2780cisR cells were used (Figure 4B), only Tofacitinib was able to slightly elevate sensitivity to RJY13. In both cell lines, Dasatinib was not efficient in restoring platinum sensitivity. Platinum resistance of OV-90 cells, while not affected by JAK/STAT inhibitors, showed significant apoptosis (high-density cPARP band) in response to Dasatinib in cells exposed to TA HS-5 CM (Figure 4C).

JAK inhibitors mildly increased RJY13 sensitivity in two of the three OC cell lines that became resistant, consequent to exposure to CM. Because the inhibition of the signal transduction pathway triggered by IL-6 yields only partial response in terms of sensitivity restoration, we speculated that other signaling pathways might be involved in mediating platinum sensitivity.

Dasatinib, a potent Abl, Scr, and c-kit inhibitor [30], exhibited weak activity in overcoming platinum chemoresistance in A2780 and A2780CisR cells mediated by CM derived from TA HS-5. In contrast, exposure to Dasatinib was potent in overriding the protective effect of CM derived from TA HS-5 in OV-90 cells. Thus, Dasatinib’s ability to override platinum chemoresistance mediated by CM derived from TA HS-5 is also specific to certain cell types.

Consistent with previous findings, Ruxolitinib also exhibited minimal effect on platinum chemoresistance in A2780CisR cells exposed to MS-5-derived CM (Figure 4D). Likewise, Ruxolitinib was unable to restore platinum chemoresistance in A2780CisR cells exposed to CM derived from differentiated 3T3-L1 (WAT) (Figure 4E).

### 2.5. Crizotinib Overcomes OC Chemoresistance Mediated by TA MSC Soluble Factors

Cytokine array data (Figure 3) show that HGF levels are significantly enriched in conditioned media (CM) that were active in promoting chemoresistance to OC cells (Figure 4). HGF is a paracrine growth factor secreted by mesenchymal cells, and it targets primarily epithelial cells [31], and it exerts its function via its receptor, HGFR or c-MET [32].

To evaluate HGF/c-MET’s role, we exposed OC cells to increasing concentrations of HGF (1 ng/mL −20 ng/mL) in the presence and absence of Crizotinib, a multi-kinase inhibitor that targets c-MET, ROS1, and ALK [33]. Cell viability in the presence of RJY13 was determined.

While results shown in Figure 5A illustrate that increasing concentrations of HGF resulted in increased proliferation and reduced platinum sensitivity, exposure to Crizotinib resulted in a significant decrease in cell counts in all samples, despite the presence of HGF at the various concentrations. Therefore, Crizotinib appears to block HGF activity in promoting platinum resistance, and it restores platinum sensitivity to OC cells, likely by blocking phosphorylation of a number of kinases, including c-MET. HGF causes a reduction in cPARP levels, consistent with reduced platinum sensitivity (Figure 5B). By contrast, combining Crizotinib with HGF (20 ng/mL) abolishes HGF’s effect (Figure 5B). A2780CisR cells exposed to RJY13 resulted in significant cPARP cleavage (Figure 5D), while exposure to Crizotinib alone resulted only in minor cleavage of cPARP. Interestingly, combination of platinum compound with Crizotinib resulted in an increase in cPARP levels significantly below the level of cPARP in cells treated with a platinum compound alone. Exposure of A2780CisR cells to CM derived from TA-HS-5 resulted in significant reduction in cPARP levels, while combination with Crizotinib partially restored platinum sensitivity to A2780CisR cells.

### 2.6. Hydroxy Estradiol (2HE2) Restores RJY13 Sensitivity to OC Cell Lines Exposed to CM Derived from MS-5 and TA HS-5

We previously reported that 2HE2 restored platinum sensitivity to OC cell lines, as mediated by direct co-culture with MSC cells, independent of ERK1/2 activation [29]. 2HE2, a precursor of 2-metoxy estradiol (2ME2), is a well-researched metabolite, which, while devoid of estrogenic activity, is beneficial in cancer therapy, as shown in clinical trials [34]. Therefore, we examined 2HE2′s ability to restore platinum sensitivity to OC cells, as mediated by secreted soluble factors from MS-5 or TA HS-5 cells. Figure 6A shows cell viability relative to platinum-untreated A2780cisR cells that were not exposed to TA HS-5 CM, and the level is designated as 1.0. Platinum treatment resulted in a significant reduction in cell viability (Figure 6). Exposure to TA HS-5 CM resulted in a significant increase in cell viability (to 0.4). As a result of 2HE2 application, proliferation was significantly reduced (by approximately 80%), restoring platinum sensitivity. Moreover, cPARP levels were reduced by about threefold in the presence of TA HS-5 CM. The combination of 2HE2 and RJY13 induced a significant increase in cPARP levels, similar to the levels of the control sample. Similarly, 2HE2 was active in restoring platinum sensitivity to A2780cisR cells exposed to MS-5-derived CM (Figure 6C). Thus, 2HE2 is active in restoring platinum sensitivity to OC cells exposed to MS-5- and TA HS-5-derived CM.

## 3. Discussion

While mesenchymal stem cells (MSCs) are known to interact with cancer cells through direct cell-to-cell contact and by secretion of paracrine factors, their exact influence on tumor progression remains unclear. Previously, we showed that direct interaction of OC cell lines with MSCs (human and murine) confers platinum chemoresistance, combined alongside ERK1/2 inhibition [19]. Moreover, a number of modulators were discovered that were active in restoring platinum sensitivity, overcoming platinum chemoresistance in ERK1/2-dependent or independent fashion [19,29]. In this study, we examined the ability of conditioned media (CM) derived from human MSCs (HS-5) or tumor-activated MSCs (HS-5) to affect OC cell lines’ platinum sensitivity. Our data show that CM derived from tumor-activated MSCs were more potent in conferring platinum chemoresistance than were CMs derived from MSC grown as a monolayer (Figure 1). The ability of CM, derived from tumor-activated MSC, to promote platinum chemoresistance was observed in several OC cancer cells. Intestinally, CM derived from MSC grown as a monolayer was potent in conferring platinum chemoresistance to OV-90 cells (Figure 1C), while the same CM was less effective in promoting platinum chemoresistance to A2780 or A2780CisR cells (Figure 1A,B). Timaner, et al. (2020) reported that MSCs may support the tumor or limit its proliferation, depending upon the tumor’s origin and differentiation status [14,35]. Therefore, it is not surprising that HS-5-derived CM did not confer chemoresistance to all OC cell lines tested, while CM derived from the tumor-activated HS-5 promoted platinum RJY13 resistance in all of the OC cell lines tested. OC “education” of the MSCs may be required for the MSCs to undergo differentiation, enabling secretion of soluble factors that promote tumorigenesis and chemoresistance.

Our results are consistent with other reports of mesenchymal stem cell conditioned medium (MSC-CM)’s ability to affect cancer cells’ sensitivity to chemotherapeutic drugs and protection from chemotherapy [36]. In addition to MSCs, adipocytes have also been reported to protect tumor cells from the effect of anticancer agents [19,37]. Additional cells that are found in ovarian TME are tumor-associated macrophages (TAMs). TAMs play important roles in multiple solid malignancies, including ovarian cancer. TAMs constitute the main population of immune cells present in the ovarian tumor microenvironment [38]. TAMs contribute to carcinogenesis, neoangiogenesis, immune-suppressive TME remodeling, cancer chemoresistance, recurrence, and metastasis. These cells are characterized by high plasticity and can be easily polarized by colony-stimulating factor-1, which is released by tumor cells, into an immunosuppressive M2-like phenotype [38]. Thus, combination of TAMs-targeting strategies with traditional treatments or immunotherapies might be a valid strategy to enhance efficacy of cancer therapy [39].

Protective effects of TME cells are not limited to chemotherapeutic agents, but they also include tyrosine kinase inhibitors (TKIs), such as Lapatinib, which target the HER2 kinase. Conditioned media derived from differentiated adipocytes increase HER2-positive breast cancer cells’ resistance to Lapatinib both in vitro and in vivo [37]. Moreover, various lipolytic inhibitors abolished adipocyte-conditioned media’s protective effect on tumor cells, suggesting a role of adipocytes in the induction of tumor cells’ resistance to TKI [37].

Our previous report [19] showed that platinum chemoresistance mediated by direct interaction of MSCs with OC cells involves ERK1/2 inactivation [19]. Furthermore, several modulators, such as fisetin [19] and 2HE2 [29], were reported to restore platinum sensitivity in an ERK1/2-dependent and independent fashion, respectively. Thus, in this study, we explored ERK1/2 phosphorylation’s role in platinum chemoresistance, as mediated by CM derived from MSC-5, differentiated 3T3-L1, and TA MSCs. Our data showed that there is no correlation between ERK1/2 phosphorylation and platinum sensitivity in OC cells exposed to CM. The CM affected ERK1/2 phosphorylation differently, depending upon the cell line. Thus, in this experimental protocol, ERK1/2 phosphorylation does not correlate to platinum chemoresistance.

The urokinase plasminogen activator receptor (uPAR) level is elevated in tumor tissue from several forms of cancer [40]. uPAR is shed from the cell surface, and its soluble form [41], soluble urokinase plasminogen activator receptor (SuPAR), has been detected in several human body fluids. High plasma levels of SuPAR in patients with colorectal cancer have been associated with poor prognosis [42]. In OC patients, it has been shown that the suPAR level is very high in ascites and serum, and it is associated with lower survival rates [41]. SuPAR depletes its ligand from the medium and prevents the initiation of cellular signaling, but it does not directly trigger signaling in the treated cells. Thus, we chose not to focus on its role in platinum chemoresistance in this study, but rather on exploring the involvement of the other two soluble factors.

IL-6 is a pro-inflammatory cytokine produced by a number of cell types, including a variety of immune cells that affect target cells such as ovarian cancer cells via the JAK/STAT pathway [43]. IL-6 can be secreted in ascites by ovarian cancer cells and tumor microenvironment cells [44]. In ovarian cancer, IL-6 is believed to be involved in host immune responses to the disease [45,46]. IL-6 signaling in ovarian cancer cells can regulate tumor cell proliferation, invasion, angiogenesis, and chemoresistance [47,48,49,50]. Lane et al. (2011) [51] reported that elevated IL-6 levels in ascites of ovarian cancer patients correlated to lower progression-free survival rates.

To explore IL-6′s role in mediated chemoresistance to OC by CM derived from TA HS-5, we utilized JAK/STAT inhibitors. Several FDA-approved drugs, targeting the JAK/STAT pathway, are available, such as Ruxolitinib, an inhibitor of JAK1 and JAK2 [52], as well as Tofacitinib, a JAK3 selective inhibitor [53].

The data presented in Figure 4 show that JAK inhibitors (Ruxolitinib and Tofacitinib) mildly restore platinum sensitivity in A2780 cells. Only Tofacinitinb was able to significantly restore platinum sensitivity in A2780CisR cells, but not in OV-90 cell lines (Figure 4). We, therefore, conclude that the response to JAK inhibitors is a cell-specific phenomenon.

Our data are consistent with other reports, demonstrating that exposure of OC cells to IL-6 promotes cisplatin resistance, accompanied by elevated pSTAT3 levels. Consequently, blocking IL-6 activity using an antibody or siRNA re-sensitized cells to platinum compounds [50]. Moreover, OC cell lines obtained from the same patient before and after development of cisplatin resistance demonstrated higher release of IL-6 after the acquisition of resistance. Furthermore, the addition of IL-6 to cell line cultures increased the levels of the cellular inhibitor of apoptosis, and it reduced cisplatin’s cytotoxic effect [50].

Furthermore, Ruxolitinib, a JAK1/2 inhibitor, was able to overcome chemoresistance and to improve survival in the immunocompetent OC mouse model system using ID8 tumor cells and MSCs, alongside a decrease in pSTAT3 levels [54]. While other JAK/STAT inhibitors were also active in overcoming JAK/STAT-dependent drug resistance [55], no data are available regarding Tofacitinib’s ability to restore platinum sensitivity to OC cells.

Jung et al. (2016) [56] described the stem cell reprogramming factor PBX1′s role in mediating chemoresistance in recurrent ovarian carcinomas. In tumor cells, ectopic expression of PBX1 promotes cancer stem cell-like phenotypes, including platinum chemoresistance. High levels of PBX1 expression in clinical samples are correlated with poor survival rates. Conversely, silencing PBX1 in platinum-resistant cells that overexpressed PBX1 sensitized them to platinum treatment [56]. Thus, the tumor microenvironment that promotes chemoresistance in OC cells is correlated to the upregulation of PBX1. Support for this hypothesis was provided by Jung et al. (2016), who showed that expression of PBX1 is under the positive control of STAT3, a transcription factor modulated by cytokines, such as IL-6 and others [56,57,58]. In addition, JAK2 inhibitors restore sensitivity to platinum resistant cells [56]. Thus, mild sensitivity to JAK inhibitors in the various cells might be mediated by regulation of PBX1 expression.

Dasatinib is a multi-kinase inhibitor, targeting Abl, Src, and c-kit, and it is an approved TKI for the treatment of CML [30,59]. Recently, we showed that Dasatinib is active in overcoming stromal platinum protection in OC cells [29]. Moreover, other reports have shown Dasatinib’s capability to override stromal-mediated chemoresistance in FLT3-positive AML cells [60]. Thus, we explored Dasatinib’s ability to overcome soluble factor(s)-mediated platinum chemoresistance in OC cells. Dasatinib exhibits weak activity in overcoming platinum chemoresistance in A2780 and A2780CisR cells. In contrast, exposure to Dasatinib was potent in overriding TA HS-5-derived CM’s protective effect in OV-90 cells. Thus, Dasatinib’s ability to override platinum chemoresistance mediated by TA HS-5-derived CM is cell type-specific, and it is similar to the activity of JAK inhibitors.

As JAK inhibitors showed limited activity in restoring platinum sensitivity to OC cells, additional signaling pathways might be involved in mediating platinum sensitivity in OC cells. Therefore, we also explored HGF’s involvement in platinum chemoresistance mediated by TA HS-5-derived CM.

Hepatocyte growth factor (HGF) or scatter factor (SF) is a paracrine cellular growth, motility, and morphogenic factor [61] that is secreted by mesenchymal cells, and it targets primarily epithelial, endothelial, and hemopoietic progenitor cells [61]. HGF’s activity is mediated by c-MET receptor tyrosine kinase [32]. The HGF/MET axis was reported as a potential key contributor to promoting chemoresistance in cancer cells [62]. In our study, exposure to increasing concentrations of HGF resulted in increased proliferation and increased platinum resistance, as evidenced by the reduction in cPARP levels (Figure 5A).

Crizotinib is a multi-kinase inhibitor targeting c-MET, ALK, and ROS [63]. Thus, we explored its ability to affect chemoresistance mediated by exogenous HGF. Our data show that the Crizotinib/HGF combination significantly decreases cell proliferation in all samples, as well as increased cPARP levels, showing that Crizotinib blocks HGF’s platinum resistance activity and restores platinum sensitivity to OC cells. These effects are likely due to inhibition of phosphorylation of a number of kinases, including c-MET. Similarly, Crizotinib was partially active in restoring sensitivity to TA HS-5-mediated OC cells (Figure 5B).

A recent report from our group showed that Crizotinib is also active in inhibiting JAK2 [64], and thus it might inhibit the activity of both IL-6 and HGF TME-derived soluble factors, which confer platinum chemoresistance. Thus, combination therapy with Crizotinib merits special attention, as it is an inhibitor of both HGF/MET signaling [63] and IL-6/JAK-STAT signaling [64], both of which are implicated in mediating TME platinum chemoresistance to OC cells. Interestingly, HGF/MET activate downstream signaling via the mitogen-activated protein kinase (ERK/MAPK) and the phosphatidylinositol 3-kinase (PI3K/AKT) pathways. Thus, combining MET/HGF inhibitors with chemotherapy, radiotherapy, or targeted therapy holds promise in overriding chemoresistance and improving chemotherapy outcomes [65]. The potential utilization of Crizotinib, a c-MET inhibitor, is relevant to the finding that, in 7–27% of OC cases, c-MET was overexpressed. Its activation was found to be related to poor prognosis in patients with high-grade serous ovarian cancer (HGSOC) [66]. In addition, our results corroborate previous findings, showing synergistic activity of Crizotinib and platinum compounds in animal models of OC [67,68].

An additional potential mode of action of HGF in mediating platinum chemoresistance might be related to regulation of miR-199a-3p, whose expression was implicated in ovarian platinum chemoresistance, and patients with low miR-199a-3p levels were more resistant to platinum, leading to a significantly poor prognosis. HGF via the MET/PI3K/AKT signaling axis downregulates miR-199a-3p expression [69].

Combining 2HE2 with RJY13 caused a significant increase in cPARP levels, similar to the levels of the control sample, suggesting that exposure to 2HE2 restores platinum sensitivity to OC cells exposed to MS-5- and TA HS-5-derived CM (Figure 6). While the exact mechanism involved in overriding TME-mediated platinum chemoresistance in OC is not yet clear, it is possible that 2HE2 also affects IL-6 and HGF signaling, which should be validated experimentally. Moreover, 2HE2 is a prodrug of 2ME2. 2-ME2 has been reported to induce cell apoptosis in K562 CML cells by downregulating anti-apoptotic protein expressions of Bcl-xl and Bcl-2 [70] and augmenting apoptosis-inducing activity of Dasatinib, an Abl/Src inhibitor [70]. Moreover, 2ME2 downregulated c-Myc gene expression that might play a role in anti-cell proliferation and inducing apoptosis. Thus, it is reasonable to speculate that 2HE2 might upregulate anti-apoptotic genes to promote apoptosis in our experimental system.

In conclusion, our study described the role of components of the tumor microenvironment active in promoting platinum chemoresistance and their suggested role in the failure of cancer therapy. We also showed that combining platinum compounds with modulators of signaling pathways involved in mediating TME chemoresistance resulted in restoring platinum sensitivity to OC cells and, in turn, overcoming platinum chemoresistance. It is of interest to translate our in vitro results to validation in pre-clinical and clinical settings toward improving cancer therapy in ovarian cancer patients.

## 4. Materials and Methods

Chemicals and reagents: Most chemicals were obtained from Sigma Aldrich Israel, Ltd. (Rehovot, Israel); otherwise, the vendor is specified. 2HE2 was obtained from Cayman Chemical (Ann Arbor, MI, USA). Ruxolitinib, Tofacitinib, Dasatinib, and Crizotinib were obtained from Selleck Chemicals, LLC (Houston, TX, USA). Recombinant human Hepatocyte Growth Factor (HGF) was obtained from Sigma Aldrich Israel, Ltd. (Rehovot, Israel). 

Cell lines: Human OV-90, a high-grade serous ovarian cancer histology type, and platinum-sensitive and resistant OC cells A2780 and A2780CisR, respectively, were obtained from the American Type Culture Collection [ATCC] (Manassas, VA, USA). The cells were cultured in RPMI 1640 complete medium supplemented with 10% (*w*/*v*) fetal bovine serum (Biological Industries, Beit Haemek, Israel), 1% (*w*/*v*) L-glutamine, 100 units/mL penicillin, and 0.1 mg/mL streptomycin. The murine adipocyte progenitor 3T3-L1 and the human and murine MSC lines HS-5 and MS-5, respectively, were maintained under the same conditions. All cell lines were grown at 37 °C in a humidified atmosphere with 5% CO_2_.

Trypan blue exclusion assay: Cells (2 × 10^5^/well) were plated in six-well plates. After 24 h, cells were treated with the specified agents. Solvent-treated samples were incubated with 0.1% (*w*/*v*) dimethyl sulfoxide. Cells were collected 72 h later, stained with 0.4% (*w*/*v*) Trypan blue solution (1:1, *v*/*v*), and counted using a hemocytometer [71].

Conditioned medium: HS-5 cells were plated at 18.7 × 10^3^ cells/cm^2^ in 75 cm^2^ cell-culture flasks and incubated overnight in 10 mL culture medium for processing within a mono-culture conditioned medium (CM). Additionally, they were placed in 5 mL of TA HS-5 (TA-HS-5) and co-cultured with OC cells, A2780, A2780CisR, or OV-90, as indicated in each experiment. The next day, the culture medium was replaced with fresh media. Mono- or TA HS-5 cultures were then incubated for three days, collected, and centrifuged at 3000 rpm for 5 min at room temperature to remove cells.

3T3-L1 differentiation: 3T3-L1 differentiation was performed, as previously described [19]. Briefly, 3T3-L1 cells were plated at 3 × 10^3^ cells/cm^2^ in DMEM, 10% FCS, 100 units/mL penicillin, 0.1 mg/mL streptomycin, and 2 mM L-glutamine. After 24 h, the medium was replaced with medium containing 1 mM dexamethasone (Dex), 5 mg/mL insulin, and 0.5 M 3-isobutyl-1-methylxanthine (IMBx). After 48 h, the medium was replaced with DMEM 5 mg/mL insulin. The last step was repeated after another 48 h. CM was collected and saved for later use as CM derived from 3T3-L1 differentiated cells. Cells were fixated and tested for differentiation using Oil Red O solution (Merck, Darmstadt, Germany). Oil red is extracted with isopropanol, and an absorbance of 492 nm was measured by spectrophotometer (Tecan, Männedorf, Switzerland).

Incubation of cells with conditioned medium (CM) experiments: Human OC cells were plated 25 × 10^3^ cells/cm^2^ in 25 cm^2^ cell-culture flasks and incubated in 2.5 mL culture medium for 24 h, after which 2.5 mL of CM were added for a final concentration of 50%. When modulators were applied, they were supplemented with the CM at the final concentrations in diluted DMSO: 6 μM Fisetin, 10 μM 2HE2, 2 μM Ruxolitinib, 2 μM Tofacitinib, 5 μM Dasatinib, and 2 μM Crizotinib. After 24 h, cells were treated with 5 μM RJY13 and incubated for an additional 24 h, following collection of cells by trypsinization. Cells were counted, then centrifuged at 3000 rpm (1000× *g*) for 5 min and washed with cold phosphate-buffered saline (PBS) twice to obtain cell pellets. Pellets were used to prepare lysates for immuno-blotting or monitoring cPARP by ELISA assay.

Hepatocyte growth factor (HGF) experiment: Human OC cells were plated at the concentration of 25 × 10^3^ cells/cm^2^ in 25 cm^2^ cell culture flasks and incubated in 2.5 mL culture medium for 24 h. Then 2.5 mL medium and hepatocyte growth factor (HGF) were added to obtain final concentrations: 1, 5, 10, and 20 ng/mL. For the Crizotinib-treated samples, the final flask concentration was 2 μM.

Immunoblotting: Immunoblotting was performed, as previously described [72,73]. Briefly, protein analysis was performed by Western blot protocol on an 8–12% acrylamide gel. Cell lysate samples were prepared for loading by adding lysis buffer (#9803 Cell Signaling Technology, Danvers, MA, USA), containing protease inhibitors (P8340 and P5726, Sigma, Germany) and a phosphatase inhibitor (P-1517, AG Scientific, San Diego, CA, USA) to the cell pellets. After 30 m, samples were centrifuged, and supernatants were tested for protein concentration using the DC™ Protein Assay (Bio Rad, Hercules, CA, USA) and absorbance at 630nm was determined. Samples (50–60 μg protein) were loaded onto the gel. Proteins were immunoblotted onto a nitrocellulose membrane (Schleicher & Schuell BioScience GmbH, Dassel, Germany), which was then blocked with 5% skim milk TBS/T and incubated with the following antibodies: anti-cleaved poly (ADP-ribose) polymerase (PARP) (Asp214) (D64E10, Cell Signaling Technology), anti-α-tubulin (sc-8035, Santa Cruz Biotechnology, Dallas, TX, USA), anti-GAPDH (#2118, Cell signaling Technology), and anti-phospho-ERK1/2 (Thr202/Tyr204) (D13.14.4E, Cell Signaling Technology), as per the manufacturers’ instructions. Secondary antibodies, HRP-linked anti-rabbit (#7074, Cell Signaling Technology), and anti-mouse (NB7539 Novus, Centennial, CO, USA) were used, as per the manufacturers’ instructions. Chemiluminescence was performed with SuperSignal™ West Pico PLUS Chemiluminescent Substrate (Thermo Fisher Scientific, MA, USA) and imaged using a HP imager. Densitometry was performed with ImageQuant v8.2 software. Some blots images were obtained using X-ray films.

Magpix magnetic beads multiplex: CM samples were diluted at 1/2, 1/5, and 1/10 ratios in universal assay buffer in duplicate, and they were tested for IL-1α, IL-1β, IL-5, IL-6, IL-8, IL-10, IL-13, IL-22, G-CSF, INF-γ, TNFα, RANTES (CCL5), leptin, and MCP-1 (CCL2) using a 96-well ProcartaPlex human multiplex kit (Thermos Fisher, USA), as per manufacturer instructions. The plate was analyzed by MAGPIX (Luminex Corporation, Austin, TX, USA) following the instrument operation procedure.

PathScan^®^ Cleaved PARP Assay: Cleaved PARP in cell lysates was measured using PathScan^®^ Cleaved PARP (Asp214) Sandwich ELISA Kit (Cell Signaling Technology, Danvers, MA, USA), as per manufacturer instructions. This ELISA kit detects cPARP of both human and murine origin.

Cytokine Array: The human XL cytokine array kit (R&D Systems, Minneapolis, MN, USA) was utilized to detect relative levels of various cytokines, chemokines, growth factors, and adipokines. Various membranes were incubated with CM collected from HS-5 and TA-HS-5. Relative to reference dots densitometry of the various dots in the two filters, the correlates to CM of HS-5 and TA HS-5 were calculated. Manhattan distance, which is the absolute sum of log 2 of fold changes and -log 10 of *p* value, was determined for high signal dots and is shown in a heatmap.

Statistical analysis: We repeated each in vitro experiment at least two to three times, showing representative data/images. Statistical analysis was performed by Student’s *t*-test, with significance values set at * *p* < 0.01 or ** *p* < 0.001.

## Figures and Tables

**Figure 1 ijms-24-07730-f001:**
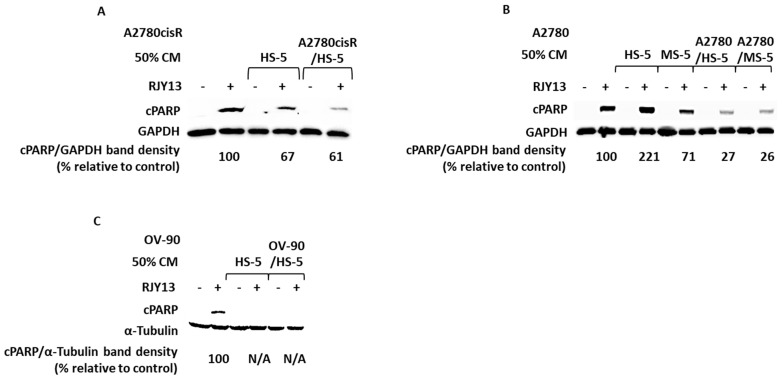
Soluble factors secreted by TA MSC confer drug resistance to OC cells. OC cell lines: A2780cisR (**A**), A2780 (**B**), and OV-90 (**C**) were exposed to 5 μM RJY13 after being pre-treated with 50% CM of MSC or 50% CM of TA-HS-5. cPARP levels were evaluated by immunoblotting. Relative to untreated, RJY13 control levels are indicated. GAPDH or α-Tubulin were used as a loading control.

**Figure 2 ijms-24-07730-f002:**
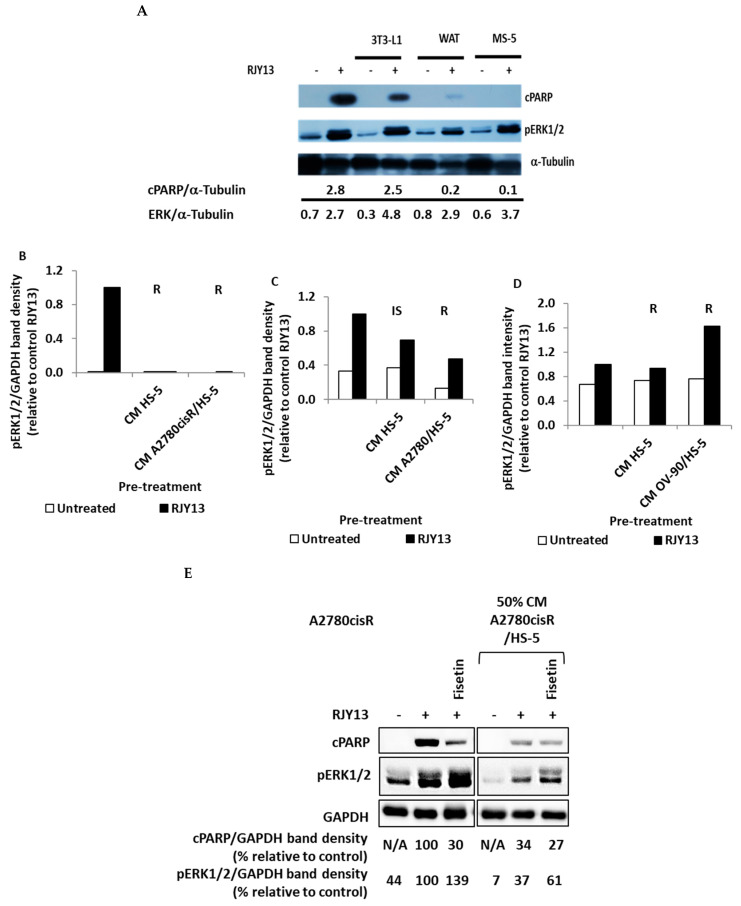
ERK1/2 phosphorylation following CM-mediated chemoresistance. OC cells A2780CisR (**A**,**B**,**E**), A2780 (**C**), and OV-90 (**D**) were exposed to 5 μM RJY13 after being pre-treated with 50% CM derived from MS-5, 3T3-L1, differentiated 3T3-L1 (WAT) (A), HS-5, or TA HS-5 (**B**–**D**). Phospho-ERK1/2 (pERK1/2) levels were evaluated by immunoblotting and quantified by densitometry relative to a housekeeping protein. Relative to the untreated sample, RJY13 control levels are indicated. GAPDH or α-Tubulin were used as a loading control. Quantitation was performed in A using Image J program.

**Figure 3 ijms-24-07730-f003:**
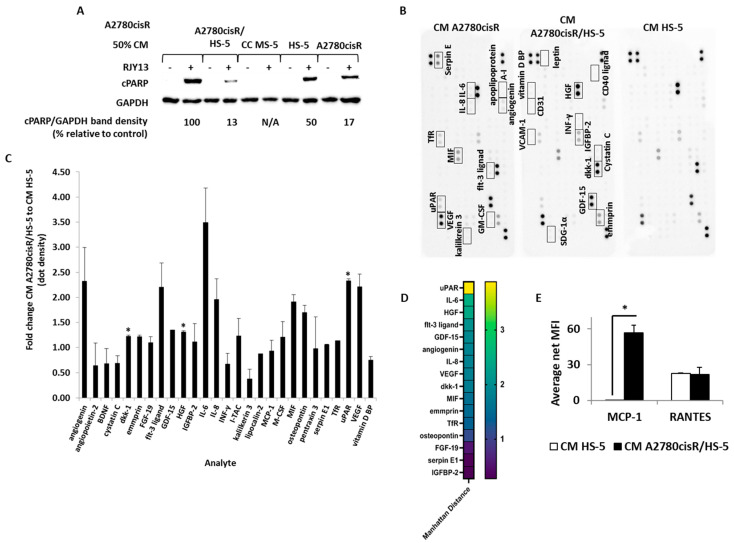
Identification of soluble factors differentially enriched in CM of TA HS-5 relative to HS-5 cells. CM collected from HS-5, A2780CisR, and TA HS-5 were aliquoted and frozen. One aliquot was used to validate the CM to promote chemoresistance (**A**). The human XL cytokine array kit (R&D Systems) was used to determine the contents of the second aliquot of each of the CMs (**B**). Dot densitometry was determined, and cytokine levels of CM in TA HS-5 to CM HS-5 ratio were calculated for detectable factors (**C**). Manhattan distance was also calculated for the appropriate ratio and ratio *p*-value (**D**) to present significant differences. In addition, costume-made multiplex ELISA (Thermo Fisher Scientific, Paisley, UK) was used to monitor enriched cytokines (IL-1α, IL-1β, IL-5, IL-6, IL-8, IL-10, IL-13, IL-22, G-CSF, INF-γ, TNFα, RANTES (CCL5), leptin, and MCP-1 (CCL2) in TA HS-5 CM, relative to HS-5 CM (**E**). Bars indicate standard deviation. IL-1β, TNFα presence. Among these, MCP-1 (CCL-2) and RANTES (CCL5) were detected in TA HS-5 CM relative to CM HS-5 levels. * *p* < 0.01.

**Figure 4 ijms-24-07730-f004:**
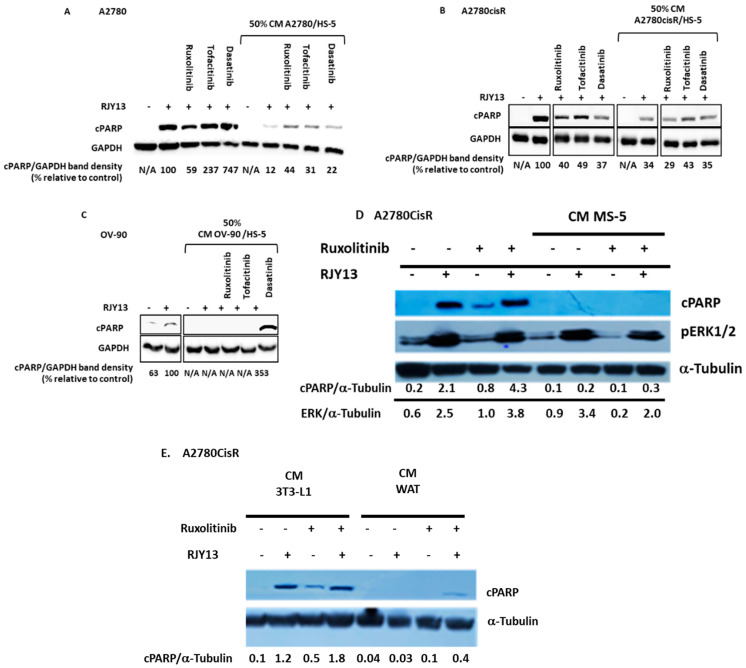
Kinase inhibitors partially restore platinum sensitivity to OC cells exposed to TA MSC. Cisplatin-responsive OC cell line A2780 (**A**), cisplatin resistant cell line A2780cisR (**B**,**D**,**E**), and OC cell line OV-90 (**C**) were exposed to 5 μM RJY13 after being pre-treated with 50% conditioned medium (CM) derived from MS-5 (**D**), 3T3-L1, and differentiated 3T3-L1 (**E**) or TA HS-5 (**A**–**C**). Additionally kinase inhibitors 2 μM Ruxolitinib, 2 μM Tofacitinib, and 5 μM Dasatinib. cPARP levels were evaluated by immunoblotting and densitometry. GAPDH or α-tubulin were used as a loading control. cPARP/GAPDH band density relative to control (%) is indicated.

**Figure 5 ijms-24-07730-f005:**
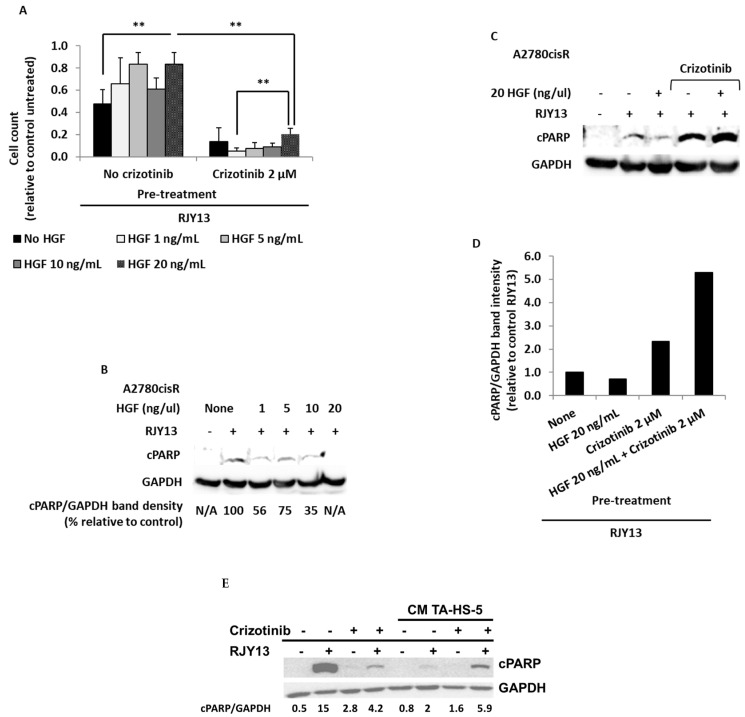
Crizotinib restores platinum sensitivity to OC cells in the presence of hepatocyte growth factor (HGF). Cisplatin-resistant OC cells, A2780cisR, were incubated with 5 μM RJY13 after being pre-treated with 1, 5, 10, or 20 ng/mL HGF with or without 2 μM Crizotinib. Cell count is presented relative to control, i.e., untreated cells (**A**). Levels of cPARP bands after, 1, 5, 10, and 20 ng/mL HGF pre-treated and RJY13 treated A2780cisR cells were evaluated by immunoblotting (**B**). A2780cisR cells were treated with RJY13 in the presence and absence of HGF (20 ng/mL) and assessed for apoptosis induction by cPARP immunoblotting (**C**). Quantitation of relative intensity of cPARP (**D**). Cisplatin-resistant A2780cisR cells were exposed to 5 μM RJY13 after being pre-treated with conditioned medium (CM TA-HS-5), as well as 2 μM Crizotinib (**E**). GAPDH was used as a loading control. Error bars indicate standard deviation. ** *p* < 0.001.

**Figure 6 ijms-24-07730-f006:**
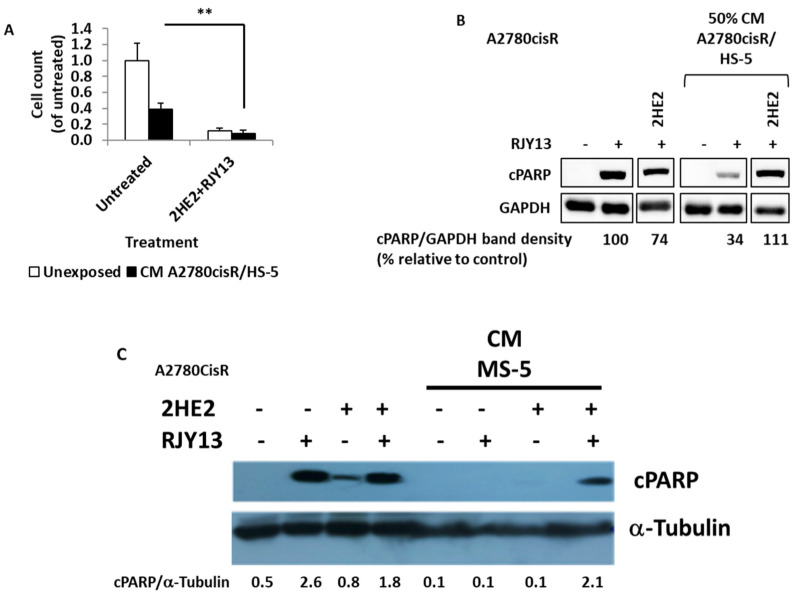
2HE2 restores RJY13 chemosensitivity to OC cells, mediated by MS-5 or TA HS-5 CM. Cisplatin-resistant A2780cisR cells were exposed to 5 μM RJY13 after being pre-treated with conditioned medium (CM) of A2780cisR/HS-5 (**A**,**B**) or MS-5 (**C**), as well as 10 μM 2HE2. Live cells were determined and presented relative to platinum-untreated cells without exposure to TA HS-5 CM, designated as 1.0 (**A**). cPARP levels were evaluated by immunoblotting (**B**,**C**). GAPDH or α-tubulin were used as a loading control. ** *p* < 0.001.

## Data Availability

The data presented in this study are available on request from the corresponding author.

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
