# Peer review of "Secreted Soluble Factors from Tumor-Activated Mesenchymal Stromal Cells Confer Platinum Chemoresistance to Ovarian Cancer Cells"

_ijms, 2023, doi:10.3390/ijms24097730_

Round 1

Reviewer 1 Report

Major concerns:

1. Most of the western blot data were overadjusted or overexposed. The internal control in some blots was not even, so the data were not convincing.

2. Some western blots did not seem to be from one gel (Fig4B and Fig6B). Were these samples from one experiment or from different experiments? 

3. Some of the western blot lack quantification (fig.5)

4. For the pERK blots, total ERK must be shown rather than tubulin

5. The efficacy of the inhibitor must be shown in Fig4 and Fig5.

Due to the poor quality of the western blot, the conclusion presented in the manuscript is not convincing enough to support the overall hypothesis of the research. A major revision is required.

Author Response

Comments are in the enclosed file

Reviewer 2 Report

The authors of this manuscript report that soluble factors from tumor-activated MSCs, including soluble uPAR, IL-6, and HGF, can confer chemoresistance to OC cells and study the related mechanism. Overall, the study is clear, and the results support the conclusion. However, there are some issues that need to be addressed to improve the quality and soundness of the manuscript. Here are some comments:

1. The abstract is too long and has redundant information. The authors should consider shortening it.

2. The tumor microenvironment consists of several cell types, including immune cells, fibroblasts, adipocytes, and mesenchymal stem cells (MSCs). However, tumor-associated macrophages (TAMs) are one of the most important components of the TME and play a critical role in regulating tumor progression and immunosuppression. TAMs also secrete factors such as IL-6 to confer chemoresistance to OC cells and downregulate immune function. The authors should discuss the functional similarities and differences between TAMs and MSCs, especially regarding the potential immune regulation function of MSCs.

3. The authors should propose strategies to reduce MSC-induced tumor chemoresistance, which is important for translational and clinical studies.

4. There are some typos: "1ng/ml -20ng/ml" should have a blank between numbers and units.

5. For the statistical bars, such as 3E and 5A, dots should be presented.

Author Response

Comments in the enclosed file

Round 2

Reviewer 1 Report

Major concerns:

1.     Most of the western blot data were overadjusted or overexposed. The internal control in some blots was not even, so the data were not convincing.

In most of the western blots shown, we utilized phospho imager and thus relative numbers of the target protein were calculated. In few cases, blots were generated from x-ray films and levels of the target protein such as cleaved cPARP were very evident even though control proteins were not even. Calculating relative levels was included in all cases using Image J.

The overadjusted image can not reflect the real results, for example: In the raw image of Fig1C, cPARP can be seen in Lane4 and Lane6, but after adjustment, the author claimed the cPARP couldn't be detected. All the WB in fig 1 were overadjusted.

2.     Some western blots did not seem to be from one gel (Fig4B and Fig6B). Were these samples from one experiment or from different experiments? 

In some cases, blots included un-relevant samples and thus were deleted them. Raw images in the relevant Fig 4B and Fig 6B are included in the file of Raw images.

I checked the raw data and the explanation was accepted

3.     Some of the western blot lack quantification (fig.5)

We added quantitation to fig 5.

 accepted

4.     For the pERK blots, total ERK must be shown rather than tubulin

We believe that pERK is largely not involved in the described phenomenon and utilizing a-tubulin is satisfactory.

 It's nothing to do with whether ERK was involved in the described phenomenon or not. Without total ERK protein control, how does the author know ERK is not involved?

5.     The efficacy of the inhibitor must be shown in Fig4 and Fig5.

Efficacy of the inhibitors are shown in Fig 4 and Fig 5.  \

Please show the inhibitor really works. The author should show the inhibitors inhibit their targets.

Due to the poor quality of the western blot, the conclusion presented in the manuscript is not convincing enough to support the overall hypothesis of the research. A major revision is required.

Although some western blots are not with high quality, never the less we provident compelling evidence for the phenomenon. The manuscript was modified and figures were improved to address most of the comments. 

Most of the comments were not well addressed

Author Response

Dear Editor,

Attached please find our response to the first reviewer.

Round 3

Reviewer 1 Report

Accept in present form